# Differences in gait parameters between children with achondroplasia and an age-matched control group of typically developed children in the age range of 6 to 12 years

Mareike Hergenröther[1]*, Katja Palm[2], Klaus Mohnike[2], Kerstin Witte[1]

**1** Sports Engineering/ Movement Science Departement, Faculty for Human Sciences, Otto-von-Guericke University, Magdeburg, Germany, **2** Children's Hospital, Otto-von-Guericke-University, Magdeburg, Germany

* mareike.hergenroether@ovgu.de

## Abstract

For people with achondroplasia, the ability to walk pain-free and longer distances is essential for maintaining independence and quality of life. However, due to shorter extremities and misalignments in the lower limbs, the walking pattern of this cohort is often affected. One possible contributing factor is an imbalanced foot-to-leg ratio. Although these structural differences are well documented, research on their specific impact on gait mechanics remains limited. In particular, no studies have investigated how an imbalanced foot-to-leg ratio, among other potential factors, may influence kinematic and kinetic parameters throughout the gait cycle. Therefore, this study examined gait differences between children with achondroplasia (N = 15) and an age-matched (6–12 y) control group (N = 15). Using a 3D motion capture system (Vicon, 13 cameras) and a modified Plug-in-Gait model, spatio-temporal, kinematic, and kinetic parameters were analyzed. Statistical parametric mapping was applied to identify key moments in the gait cycle where significant deviations occurred between the two cohorts. Results showed that all spatio-temporal parameters, except cadence, differed significantly between the groups (p < 0.05). Most kinematic and kinetic parameters also showed significant differences, particularly around initial contact and toe-off. Notably, kinematic deviations were found at the pelvis, hip, and ankle during initial contact. Around toe-off, ankle kinematics were significantly different (p = 0.006). Kinetic differences at the hip and knee in both planes were also evident around toe-off. These findings suggest that an imbalanced foot-to-leg ratio may influence gait patterns, particularly during initial contact and toe-off phases. However, additional factors contributing to these deviations remain to be identified. Further research is necessary to elucidate these influences, which could support the development of more targeted and individualized therapeutic interventions for individuals with ACH.

**Data availability statement:** All relevant data are within the manuscript and its Supporting Information files.

**Funding:** The author(s) received no specific funding for this work.

**Competing interests:** The authors have declared that no competing interests exist.

## Introduction

Achondroplasia (ACH) is a form of disproportionate short stature, occurring in approximately 1 in 25,000–30,000 births worldwide [1]. It is caused by a mutation in the FGFR3 (Fibroblast Growth Factor Receptor 3) gene, which primarily affects the growth of the extremities, while sitting height remains within the lower normal range [2]. Characteristic features of ACH include an increased anterior pelvic tilt and leg malalignments, often leading to either crossed or bowed legs, with a stronger tendency toward bowing [1,2]. This bowing is further attributed to greater growth impairment in the femur compared to the tibia [2].

Individuals with ACH commonly experience difficulties walking long distances [3] and exhibit gait deviations that impose excessive mechanical loads on the lower extremity joints [4]. A three-dimensional gait analysis (3DGA) is a valuable tool for gaining a detailed understanding of an individual's gait pattern. However, research on ACH gait using 3DGA is limited [4–9]. Existing studies consistently report key characteristics such as an increased anterior pelvic tilt and an increased flexion pattern of the hip, knee, and ankle, as well as increased knee valgus and/or varus. The observed flexed gait pattern has been hypothesized to result from the relatively longer foot length in proportion to shorter legs, due to imbalanced growth between the femur and tibia [5,8,9]. However, this hypothesis remains unverified.

Regarding kinetic parameters, research remains scarce, and the available findings are inconsistent [4,5]. Some studies have reported larger internal knee valgus moments and increased hip abduction moments in individuals with ACH [4,5]. Additionally, most existing literature focuses on overall differences between ACH and control groups by averaging gait cycles (GC) and analyzing peak values of selected parameters. An exception is the study by Kiernan [6], which employed statistical parametric mapping (SPM) [10]. SPM enables the identification of specific points within a GC where significant deviations occur, providing a more detailed understanding of gait abnormalities.

Literature suggests that the altered foot-to-leg ratio in ACH may contribute to gait deviations, particularly influencing flexion patterns [8,9]. Therefore, this study hypothesizes that differences in foot-to-leg ratio between children with ACH and age-matched children of average height (CAH) will significantly affect spatiotemporal gait parameters. As a result, it is expected that kinematic and kinetic gait parameters will be altered, particularly at initial contact (IC) and toe-off (TO). While gait patterns are influenced by factors such as age, reduced stature, and mobility adaptations, our study differs from Broström et al. [5] by narrowing the age range to 6–12 years and analyzing gait deviations throughout the entire gait cycle rather than focusing solely on peak angles and moments.

This study aims to provide a more comprehensive analysis of spatiotemporal, kinematic, and kinetic gait parameters in children with ACH compared to CAH. It is hypothesized that children with ACH will exhibit significant deviations in these parameters, particularly at IC and TO, which may be associated with their altered foot-to-leg ratio. The findings of this study will contribute to a more detailed understanding of ACH gait patterns, which may aid in aiding the development of improved therapeutic and intervention strategies.

## Methods

### Participants

A total of 30 children (15 ACH and 15 CAH) aged 6–12 years participated in this cross-sectional study (Table 1). Children with ACH were under regular care at the University Children's Hospital. Medical check-ups confirmed that none of the participants with ACH showed signs of neuropathy caused by spinal canal abnormalities. All children in the ACH group were undergoing treatment with Vosoritide, a medication aimed at increasing growth velocity (see Table in S1 Table for the current duration of treatment).

For the control group, children were recruited from local schools with various backgrounds across the city. Invitations to participate were sent to the legal guardians of children in the appropriate age range. The sample size for the ACH group was determined based on the number of children regularly receiving treatment at the University Children's Hospital. For the CAH group, 15 children within the specified age range were randomly selected.

Inclusion criteria for both groups required participants to be between 6 and 12 years of age, with no history of previous surgery or acute injuries affecting the lower limbs, and no malalignments in the knee or ankle regions for the CAH group. All participants needed to be capable of standing and walking independently, without any assistive devices.

Before participation, children and their legal guardians were orally briefed on the study procedures, and all questions were addressed. Written informed consent was obtained from both the participants and their legal guardians. This study was approved by the Ethics Board of Otto-von-Guericke University Magdeburg (Approval No. 82/23) and was conducted in accordance with the Declaration of Helsinki. Participant recruitment occurred between August 1, 2023, and August 28, 2024.

### Protocol

Data collection took place at the sports department of the authors' university, utilizing a 13-camera Vicon system with a sampling frequency of 200 Hz. The following variables were recorded: age (years), height (cm), body mass (kg), sitting height (cm), the standard deviation scores of the body mass index ($BMI\text{-}SDS_{LMS}$), foot-to-leg ratio, and standing-to-sitting ratio (Table 1). The foot-to-leg ratio was defined as the ratio of leg length to foot length, expressed as a percentage. Similarly, the standing-to-sitting height ratio was calculated by dividing standing height by sitting height. BMI-SDS was calculated following the equation of [11].

$$BMI - SDS_{LMS} = \frac{\left[\frac{BMI}{M(t)})\right] *^{L(t)} -1}{L(t)S(t)}$$

(1)

**Table 1. Anthropometric data is presented as median and range. P-value set to 0.05. Effect size is calculated as pearson correlation coefficient between ACH and CAH.**

|  | ACH (n = 15) | CAH (n = 15) | p value | Effect size |
|---|---|---|---|---|
|  | Median (Range) | Median (Range) |  | (r) |
| **Sex** | m = 9/ f = 6 | m = 11/ f = 4 |  |  |
| **Age (years)** | 10.25 (6.46-12.81) | 10.25 (6.54-14.26) | 0.756 | 0.06 |
| **Height (cm)** | 112.8 (100.3-127) | 140.6 (121.1-157) | <.001 | −0.83 |
| **Weight (kg)** | 28.7 (21.8-47.7) | 32.2 (22.7-29.2) | 0.164 | −0.25 |
| **BMI – SDS** | 0.44 (−2.11–2.85) | −0.1 (−1.60-1.40) | 0.081 | 0.32 |
| **Sitting Height (cm)** | 73.5 (62-85) | 73.5 (65-82) | 0.775 | 0.06 |
| **Foot-Leg Ratio (%)** | 40.5 (36-46) | 32 (28-33) | <.001 | 0.85 |
| **Standing-Sitting Ratio (%)** | 66 (61-69) | 52 (50-57) | <.001 | 0.86 |

whereas BMI is the individual score, and t is the age of the child. $L(t)$ is the box-cox transformation, $M(t)$ the median and $S(t)$ the coefficient of variation [11].

Prior to the data collection, participants were queried about their current well-being and then provided with oral instructions regarding the task. After anthropometric measurements were taken, 28 retroreflective markers were applied to specific anatomical landmarks by an experienced researcher, following a modified plug-in gait model [12]. A static calibration was then performed using Vicon Nexus System 2.12.1 (Vicon Motion Systems, United Kingdom).

Participants were instructed to walk multiple times along an approximately 10-meter walking path at their self-selected walking speed. The trials were recorded using the Vicon Nexus software, and ground reaction forces were measured with two in-ground integrated force plates (AMTI MiniAMP MSA-6, AMTI, Watertown, USA) at a sampling frequency of 1000 Hz. Three valid kinetic trials for each leg were targeted to allow for the analysis of mean values for the selected parameters.

## Data processing

Raw trials were processed using specific pipelines in Vicon Nexus and exported as CSV files. Spatio-temporal parameters were extracted from Vicon Nexus, while kinematic and kinetic parameters were analyzed in Vicon Polygon. These parameters were time-normalized, and the average of six valid kinetic trials (three for the left leg and three for the right leg) was calculated for each participant. The average was used after confirming symmetry between the left and right legs for each individual. The spatio-temporal parameters of interest included walking speed, step length, and cadence, as well as their normalized versions. Kinematic parameters of interest included hip flexion/extension, hip adduction/abduction, knee flexion/extension, knee varus/valgus, ankle dorsiflexion/plantarflexion, ankle inversion/eversion, pelvic tilt, and pelvic obliquity. Kinetic parameters were similar to the kinematic parameters, with the exception of the pelvic parameters.

The transverse plane was excluded from the analysis due to the specific location of the hip joint center (HJC) in individuals with ACH, which cannot be properly calculated by the software algorithms.

Spatio-temporal parameters were normalized by leg length, following the protocol of Hof [13], while kinetic parameters were normalized by body weight. All kinematic and kinetic curves were resampled to 100 samples per GC.

## Statistics

Descriptive statistics (median and range) were used to summarize the anthropometric parameters. The data were tested for normality using the Shapiro-Wilk test. For anthropometric and spatio-temporal parameters, the Mann-Whitney U test was applied.

For differences in the kinematic and kinetic parameters between the groups, Statistical Non-Parametric Mapping (SnPM) was used (SPM1d version 0.4, available for download at http://www.spm1d.org/) in MATLAB (The MathWorks Inc., Natick, MA, 2015). SnPM provides the ability to detect significant differences for each data point within a GC between two cohorts, thereby allowing the identification of more specific differences in the gait patterns. Originally developed for brain research, SPM has been widely adopted in biomechanics and is recommended for hypothesis testing with one-dimensional data [14]. It has already been applied to study gait patterns and detect abnormalities in pathological cohorts, including individuals with cerebral palsy [15–17] and Duchenne muscular dystrophy [18].

Given the small cohort and the individuality of the children, the non-parametric t-test was chosen for the analysis. The alpha level was set at 0.05. Significant differences are indicated as shaded gray areas in the graphs of the SnPM results.

Pearson correlation (r) was used to investigate the relationship between the foot-to-leg ratio and selected kinematic and kinetic parameters at the events of IC and TO. The effect size for anthropometric, kinematic and kinetic parameters, Pearson correlation was used, with the following effect size thresholds: small effect: 0.1, medium effect: 0.3, and large effect: 0.5.

 

## Results

### Anthropometric

Using the Mann-Whitney U test, the known characteristics ACH were clearly distinguishable (Table 1). Significant differences were observed for body height (U = 3.500, Z = −4.522, p < 0.001, r = −0.83), foot-to-leg ratio (U = 225.000, Z = 4.673, p < 0.001, r = 0.85), and standing-to-sitting height ratio (U = 225.000, Z = 4.690, p < 0.001, r = 0.86).

### Spatiotemporal parameters

Significant differences were found between ACH and CAH for several spatiotemporal parameters (Table 2). Significant differences were noted for walking speed (m/s) (U = 41.000, Z = −2.97, p = 0.002, r = −0.54) and step length (m) (U = 4.000, Z = −4.502, p < 0.001, r = −0.82).

When normalized by leg length, normalized cadence (U = 38.000, Z = −3.090, p = 0.002, r = −0.56), and normalized step length (U = 182.00, Z = 2.88, p = 0.003, r = 0.53) exhibited significant differences. However, cadence (steps per minute) (U = 158.00, Z = 1.89, p = 0.061, r = 0.34) did not show a significant difference, though it did exhibit a medium effect size.

### Kinematics

**Sagittal.** Significant differences between ACH and CAH were observed in joint kinematics in the sagittal plane for pelvic, hip, and ankle parameters (Figs 1–3). The pelvic tilt was significantly increased throughout the entire GC (p = 0.001). Additionally, the ankle angle showed significant differences at IC (p = 0.018) and TO (p = 0.006). The hip angle exhibited significant differences around 10% of the GC (p = 0.021), during the mid-stance phase (p = 0.001), and in the terminal swing phase (p = 0.009). However, knee flexion/extension did not reach statistical significance.

**Frontal.** Significant differences were observed in pelvic, hip, and ankle angles in the frontal plane across the GC (Figs 1–3). The ankle angle was related to the pelvic angle in the sagittal plane, showing significant differences throughout much of the GC: from 0–5% (p = 0.013), 5–60% (p = 0.001), 75–85% (p = 0.006), and 95–100% (p = 0.011). The hip angle also showed significant differences during the stance phase, including at IC (0–10%, p = 0.003) and near the end of the GC (~100%, p = 0.021). Lastly, the pelvic angle demonstrated significant differences around 40% of the GC (p = 0.011) and in the terminal swing phase (p = 0.005). Knee varus/valgus did not show statistical significance.

### Kinetics

**Sagittal.** In the sagittal plane, significant differences in kinetic parameters, presented as moments, were observed at specific points of the GC (Figs 4 and 5). The hip flexion/extension moment showed significant differences at several points

Table 2. Spatio-temporal parameters are presented as median and range. P-value set to alpha = 0.05. The right side of the table is the comparison to the parameters of Broström et al. [5].

| Current Study | | | | | | Broström et al. 2022 | | |
|---|---|---|---|---|---|---|---|---|
| | ACH (n = 15) | | CAH (n = 15) | | p value | ACH (n = 16) | CAH (n = 19) | p value |
| | Median (Range) | | Median (Range) | | | Mean (SD) | Mean (SD) | |
| Walking speed (m/s) | 0.93 | (0.67-1.32) | 1.27 | (0.77-1.69) | **0.002** | 0.98 (0.20) | 1.32 (0.15) | **0.000** |
| Step length (m) | 0.39 | (0.31-0.50) | 0.58 | (0.47-0.68) | **<0.001** | 0.38 (0.06) | 0.61 (0.08) | **0.000** |
| Cadence (steps/min) | 140.77 | (121.56-178.52) | 132.18 | (97.66-189.65) | 0.061 | 152.8 (21.0) | 128.3 (14.5) | **0.000** |
| Normalized walking speed | 0.45 | (0.34-0.64) | 0.49 | (0.27-0.56) | 0.638 | 0.49 (0.10) | 0.48 (0.06) | 0.977 |
| Normalized cadence | 0.31 | (0.25-0.37) | 0.35 | (0.28-0.41) | **0.002** | 19.1 (2.8) | 12.9 (1.6) | **0.000** |
| Normalized step length | 0.94 | (0.68-1.05) | 0.78 | (0.58-0.94) | **0.003** | 0.90 (0.13) | 0.79 (0.09) | **0.004** |

SD = Standard Deviation.

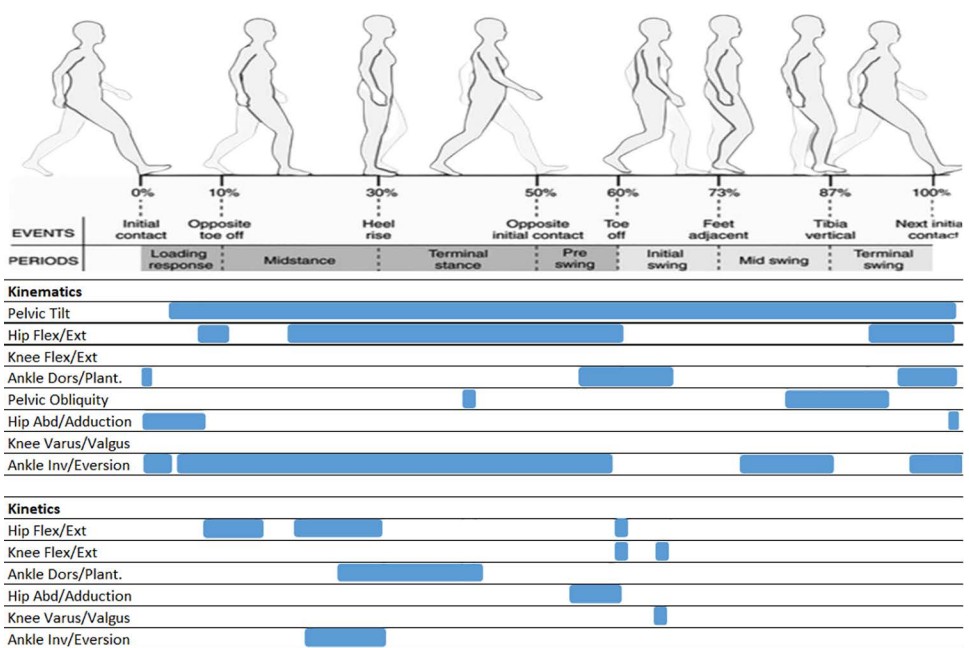

**Fig 1. Presentation of significant differences of the kinematic and kinetic parameters during GC (ACH vs. CAH). Blue bars displaying significant differences.**

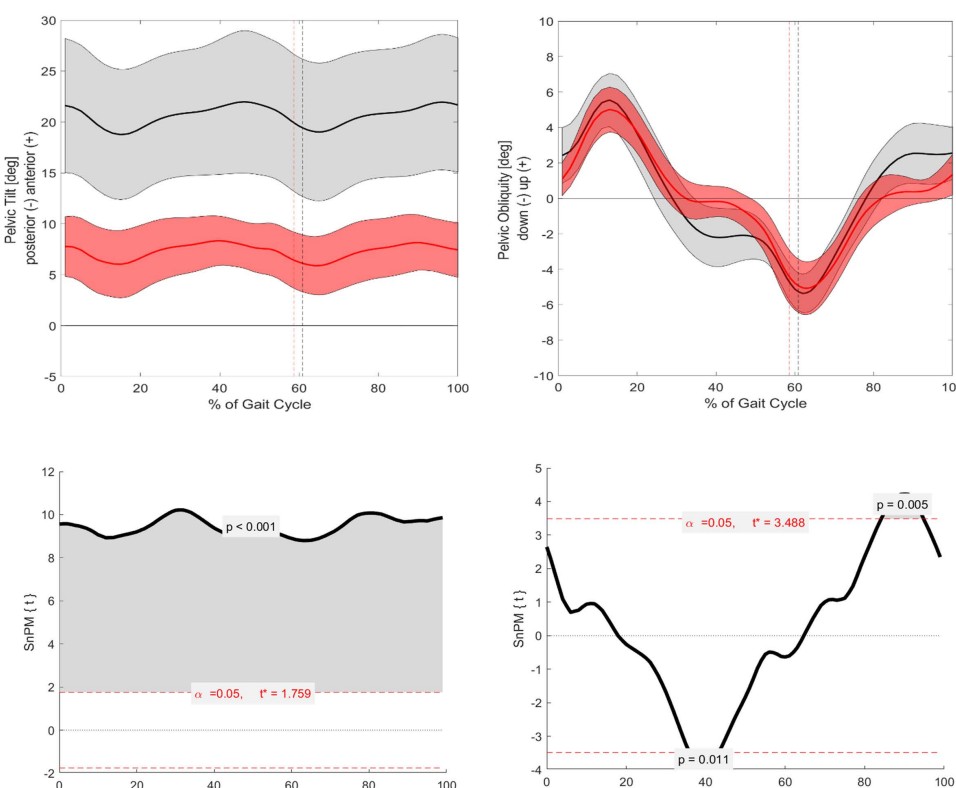

**Fig 2. Visualization of the kinematic parameters in the sagittal and frontal plane for the pelvic.** Furthermore, the results of the SnPM. Alpha level set to 0.05. Black (Grey) = ACH (SD); Red (Light red) = CAH (SD). SnPM: The grey area is visualizing the significant differences.

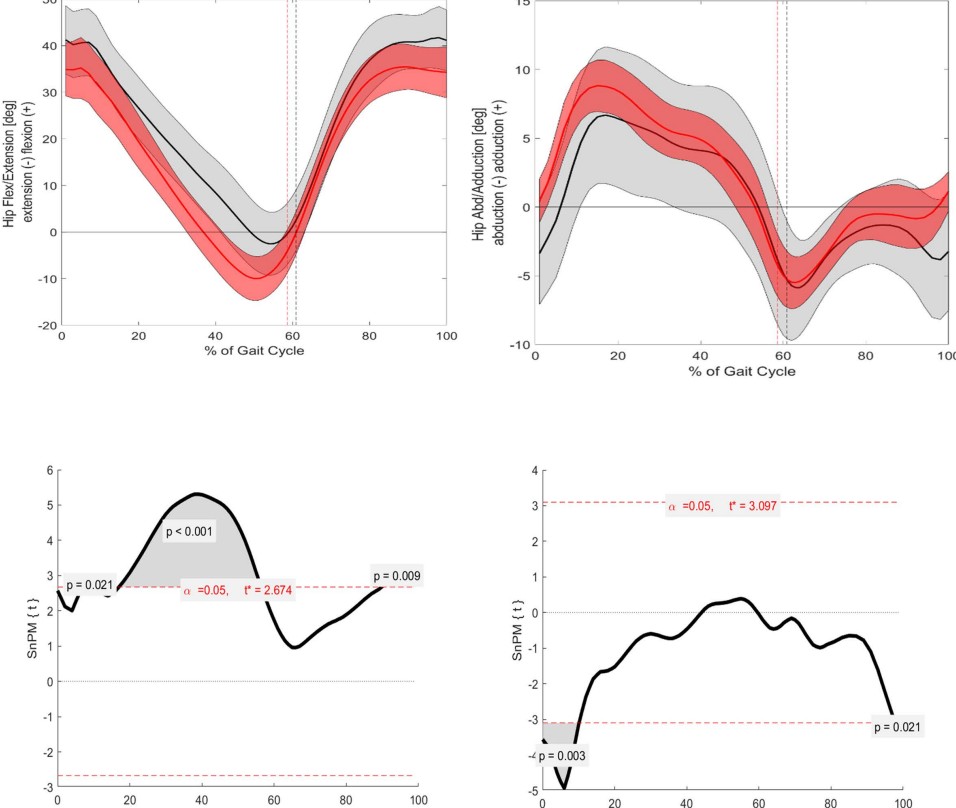

**Fig 3. Visualization of the kinematic parameters in the sagittal and frontal plane for the hip.** Furthermore, the results of the SnPM. Alpha level set to 0.05. Black (Grey) = ACH (SD); Red (Light red) = CAH (SD. SnPM: The grey area is visualizing the significant differences.

during the stance phase: 10–15% (p = 0.001), 20–30% (p = 0.003), and around 60% (p = 0.001). The knee flexion/extension moment exhibited significant differences around TO (p = 0.001). Finally, the ankle dorsiflexion/plantarflexion moment demonstrated a significant difference around the mid-stance phase (p = 0.001).

Frontal.  In the frontal plane, all kinetic parameters also exhibited significant differences (Figs 4 and 5). The hip adduction/abduction moment showed significant differences around TO (p = 0.003). The knee adduction/abduction moment displayed significant differences at 55–60% (p = 0.001) and around 65% of the GC (p = 0.002). Finally, the ankle inversion/eversion moment demonstrated significant differences at the beginning of mid-stance (p = 0.003).

## Discussion

### Anthropometric parameters

The differences observed in this study align with those reported in the literature [1,2,5–8]. As expected, ACH exhibited a significantly shorter body height, as well as a greater foot-to-leg and standing-to-sitting height ratio compared to CAH. The increased foot-to-leg ratio further supports the hypothesis, as proposed by Broström et al. [5], Kiernan [6], and Sims et al. [7], that children with ACH will exhibit an altered gait pattern due to this change in foot-to-leg ratio.

Regarding BMI-SDS, this study calculated the index for CAH according to the equation of Kromeyer et al. [11], and for ACH, the data from Neumeyer et al. [19] were used (see Equation 1). BMI-SDS is a more suitable measure than BMI for assessing the changing body composition of children, especially given that Neumeyer et al. [19] specifically developed

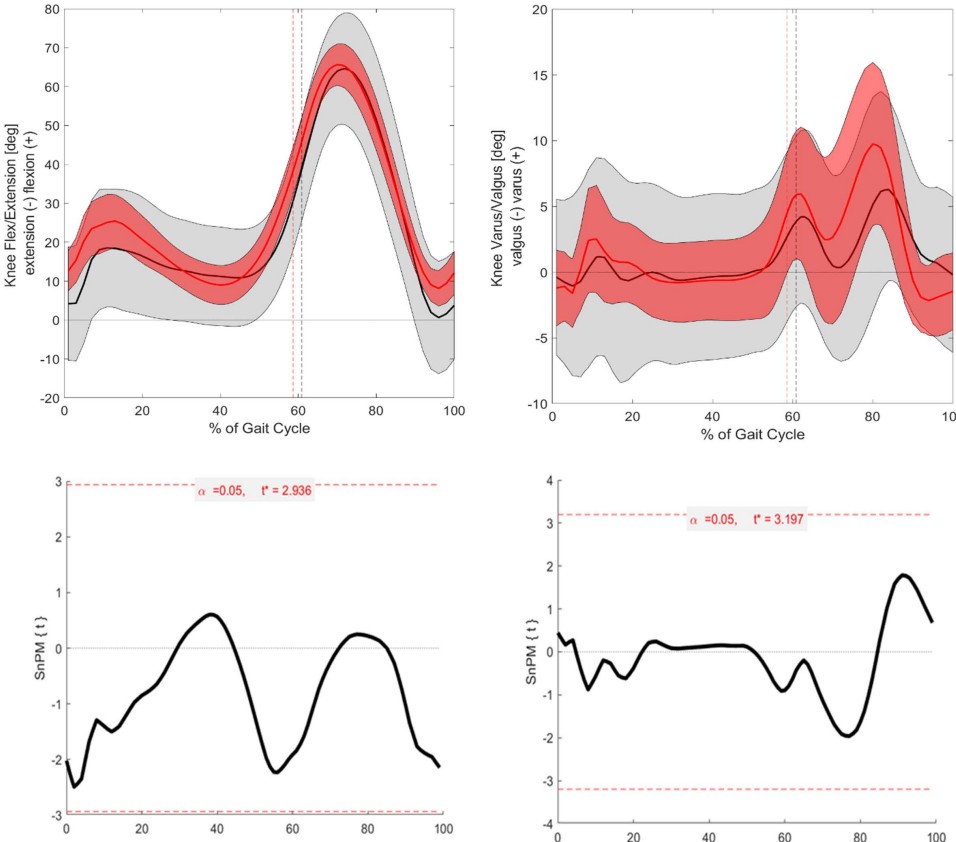

**Fig 4. Visualization of the kinematic parameters in the sagittal and frontal plane for the knee.** Furthermore, the results of the SnPM. Alpha level set to 0.05. Black (Grey) = ACH (SD); Red (Light red) = CAH (SDSnPM: The grey area is visualizing the significant differences.

their data for the ACH cohort, making it more accurate for this population. Despite these considerations, no significant differences were found between the cohorts concerning BMI-SDS.

## Spatio-temporal parameters

The results of the Mann-Whitney U test for spatio-temporal parameters show significant differences for all parameters, except for cadence and normalized walking speed (Table 2). When comparing these findings to those of Broström et al. [5] and Sims et al. [8], there is partial agreement. The walking speed results align with Broström et al. [5] (ACH: 0.98 (0.79–1.21), CAH: 1.33 (1.14–1.62)) and Sims et al. [8] (ACH: 1.02, CAH: 1.33), whereas in this study, the values were ACH: 0.93 (0.67–1.32) and CAH: 1.27 (0.77–1.69).

Furthermore, the cadence and step length parameters were consistent with those reported by Broström et al. [5]. The normalized parameters are only comparable to Broström et al. [5] due to the differences in normalization methods and desired output parameters (Table 2). Broström et al. [5] presented their results as means and standard deviations, whereas this study used the median and range. The normalized walking speed in this study aligns with Broström et al. [5], but the normalized cadence presents a smaller value in this work. However, like in Broström et al. [5], a significant difference was detected between the cohorts. Finally, the normalized step length in both studies shows equally significant difference.

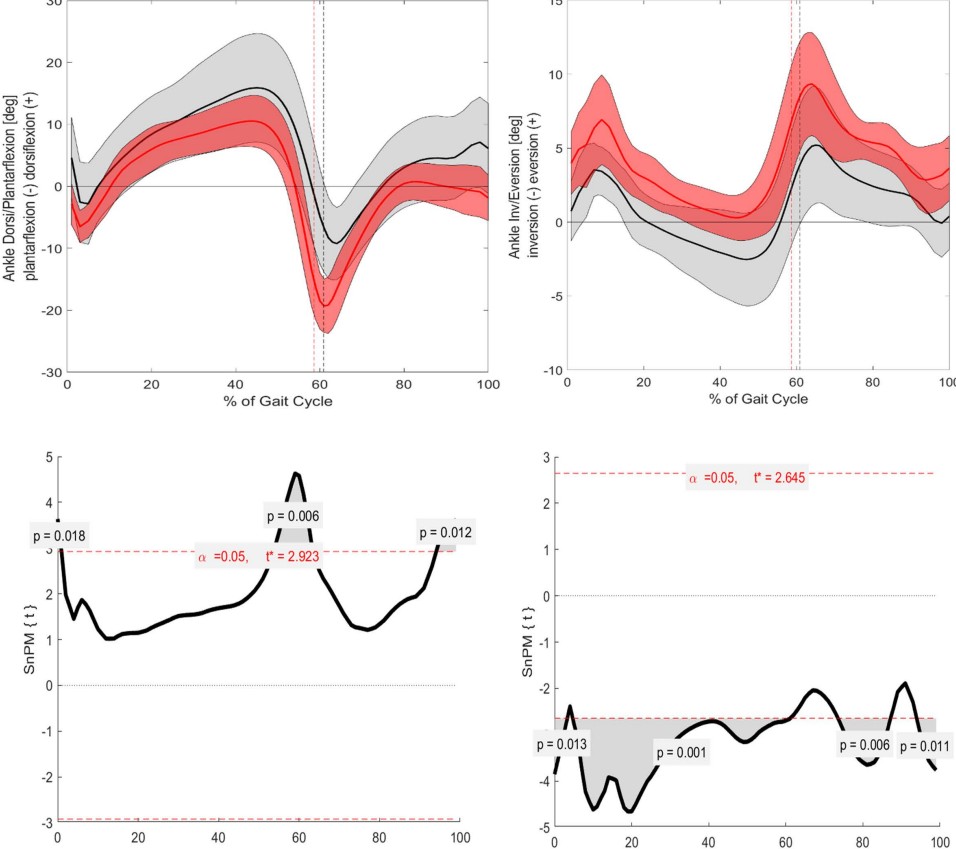

**Fig 5. Visualization of the kinematic parameters in the sagittal and frontal plane for the ankle.** Furthermore, the results of the SnPM. Alpha level set to 0.05. Black (Grey) = ACH (SD); Red (Light red) = CAH (SDSnPM: The grey area is visualizing the significant differences.

The discrepancies in normalized cadence may be explained by an issue with the application of the Hof [13] equations for normalization. In contrast, the results of this study align well with Schwartz et al. [20,21], as well as those of Broström et al. [5], with the exception of the normalized cadence (ACH: 19.1 (2.8), Control group: 12.9 (1.6))

## Kinematic parameters

**Sagittal.** Building upon existing literature, which identifies common gait characteristics in individuals with ACH, such as an anteriorly shifted pelvic tilt, increased flexion at the hip, knee, and ankle, and heightened knee valgus/varus [4–8], this study confirms most of these known findings (Fig 1). This work investigates the four key angles of the lower extremities: pelvic, hip, knee, and ankle. In the sagittal plane, significant differences are observed for three of these angles (Figs 1–4).

Specifically, for the ankle angle, significant differences are noted around the events of IC and TO, which aligns with the hypothesis of this study. The described flexion pattern [5,8] is evident in the visual standard deviations (SD) in the graphs for the ankle and hip angles in the ACH cohort. Additionally, the flexion pattern is further underscored during IC and TO, reinforcing the notion that individuals with ACH exhibit a higher flexion pattern to lift their feet off the ground.

The hip angle is significantly more flexed during the majority of the mid-stance and terminal stance phases, which contrasts with the control group's ability to achieve greater extension during these phases of the GC.

**Frontal.** In the frontal plane, pelvic obliquity in the ACH cohort significantly decreases by around 40% of the GC, corresponding to the terminal stance phase. Additionally, a significantly increased pelvic obliquity is observed at the end of mid-swing and throughout the terminal swing phase, which aligns with findings from Broström et al. [5] and Kiernan [6].

For the hip abduction/adduction angle, significant differences are present at 0–10% and around 100% of the GC, with ACH showing a tendency toward greater abduction movement during these phases. This supports the study's hypothesis that gait deviations would primarily occur around IC and TO.

Interestingly, the knee valgus/varus angle does not show statistical significance between the cohorts, despite previous literature [4–6,8] documenting differences. A possible explanation is the heterogeneity of misalignments in individuals with ACH and the averaging method used to represent both cohorts' data. Additionally, despite the exclusion criteria, natural growth variations in this age range may lead to minor misalignments in the control group, as orthopedic consultations were not conducted.

Lastly, the ankle angle demonstrates significant differences over most of the GC, with ACH consistently showing greater inversion. This increased inversion may be a compensatory mechanism due to the commonly observed genu varum, which forces the feet into a more internally rotated position.

## Kinetic

In this study, the kinetic parameters were analyzed by examining the joint moments of the hip, knee, and ankle, given the focus on the lower extremities. The decision was made to investigate joint moments as they provide insight into the mechanical demands placed on each joint during gait. To ensure comparability between participants, all kinetic parameters were normalized to body weight.

**Sagittal.** In the sagittal plane, significant differences are observed for all three joint moments, specifically for the hip, knee, and ankle. These differences are particularly noticeable around the TO, where ACH presents a distinct kinetic pattern compared to CAH (Fig 1 and Figs 6–8). The differences around TO may be related to the delayed push-off observed in ACH, as previously reported in the literature [8]. This delay could explain why the decrease in joint moment is more gradual in ACH compared to CAH. However, significant differences are also found at other phases of the GC.

For the hip flexion/extension moment, significant differences occur at the beginning of the stance phase (between 10% and 15% of the GC) and during mid-stance. At these points, ACH exhibits a larger flexion moment in comparison to CAH. In contrast, CAH shows a strong peak in the flexion moment at around 10–15% of the GC, which aligns with findings from Schwartz et al. [20].This suggests that ACH may have difficulty generating the same peak moment at this phase of the gait cycle. Furthermore, the hip flexion moment remains significantly greater in ACH at TO, further supporting the hypothesis that ACH presents a delayed and altered push-off strategy.

For the knee flexion/extension moment, significant differences are also observed, particularly at the TO event, where ACH presents a larger flexion moment compared to CAH. During the mid-stance phase, ACH shows less knee extension than CAH, although this difference does not reach statistical significance. However, in the terminal stance phase, ACH demonstrates a larger knee extension moment in comparison to CAH. This suggests that ACH may rely on increased knee extension moments to compensate for their altered gait mechanics.

For the ankle dorsiflexion/plantarflexion moment, ACH is unable to generate the same moment as CAH throughout the entire mid-stance phase. As a result, significantly lower moments are observed at certain phases of the GC. This difference is likely related to the altered foot-leg ratio in ACH, which affects the moment arm and limits their ability to generate comparable plantarflexion moments. These biomechanical constraints could explain the observed reductions in ankle joint moment in ACH throughout mid-stance.

**Frontal.** Building on the results observed in the sagittal plane, significant differences were also found in the frontal plane for all three joint moments. Notably, the hip and knee moments exhibited significant differences around the TO (Figs 1, 6, and 7). This finding supports the hypothesis that differences between the groups would be most pronounced around

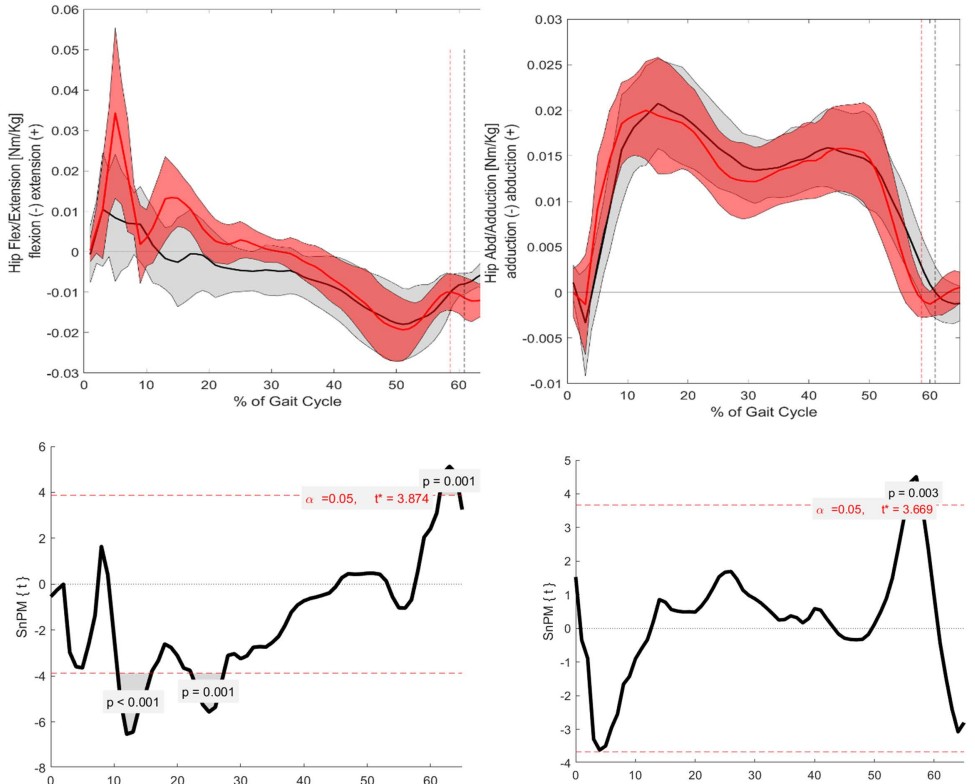

**Fig 6. Visualization of the kinetic parameters in the sagittal and frontal plane for the hip.** Furthermore, the results of the SnPM. Alpha level set to 0.05. Black (Grey) = ACH (SD); Red (Light red) = CAH (SD). SnPM: The grey area is visualizing the significant differences.

TO. Additionally, the larger hip abduction moment observed around TO aligns with findings reported in previous literature [4,5].

Beyond these significant differences, further non-significant trends were also evident. For the hip abduction/adduction moment, a larger adduction moment was observed in ACH during the middle of the loading response phase. This trend is consistent with the kinematic findings for hip abduction/adduction. Similarly, the knee varus/valgus moment followed a pattern similar to the knee flexion/extension moment. During mid-stance, ACH exhibited a larger varus moment, whereas in the terminal stance phase, a larger valgus moment was present compared to CAH.

Finally, the ankle inversion/eversion moment showed a larger inversion moment for ACH during mid-stance. This observation aligns with the kinematic data, which revealed a consistently larger inversion angle throughout the GC for ACH. Consequently, this increased inversion likely results in a greater load on the lateral side of the ankle in ACH.

### General discussion

SPM is widely used to analyze and understand movement patterns in greater detail, supporting the development of targeted therapies and interventions [14,15]. The findings of this study reinforce the value of SPM in gait analysis, highlighting its potential to identify deviations and impairments with increased precision. However, selecting the most appropriate statistical test is crucial when applying SPM. Given the heterogeneity and small sample size of the cohorts in this study, a non-parametric SnPM t-test was chosen.

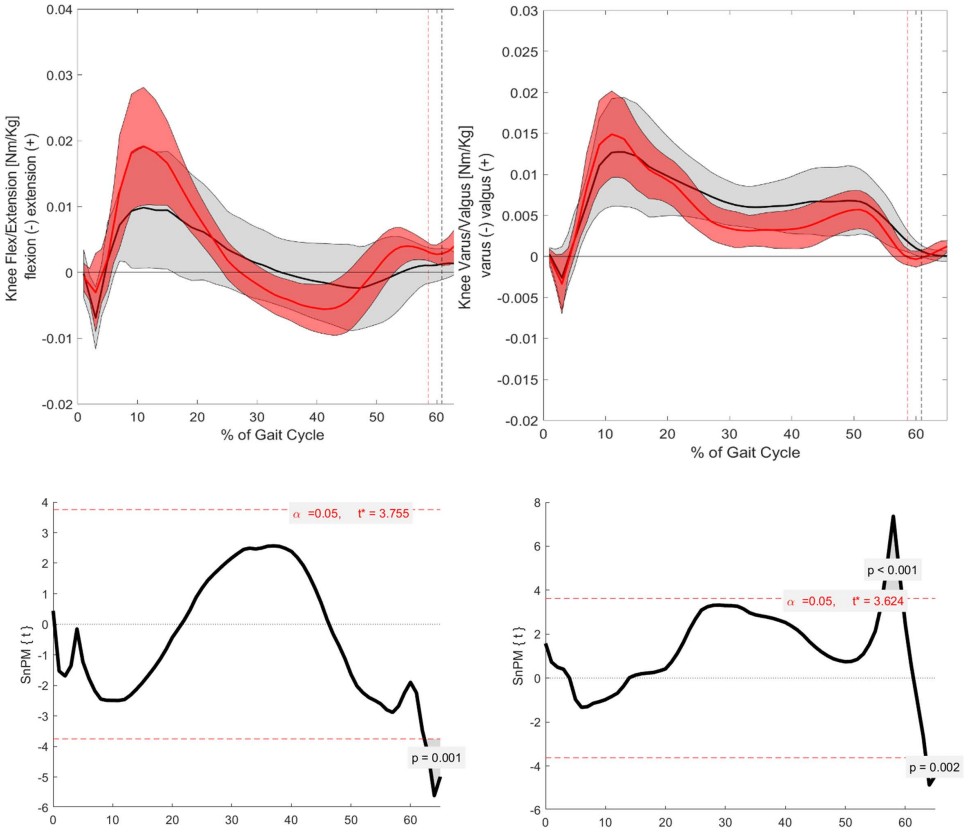

**Fig 7. Visualization of the kinetic parameters in the sagittal and frontal plane for the knee.** Furthermore, the results of the SnPM. Alpha level set to 0.05. Black (Grey) = ACH (SD); Red (Light red) = CAH (SD). SnPM: The grey area is visualizing the significant differences.

Despite narrowing the age range to 6–12 years, compared to previous studies [4–8], this period still represents a phase of stable growth velocity. The natural variations in growth development are evident in the data, as reflected in the observed fluctuations within the graphs. All spatiotemporal parameters, except cadence, showed significant differences accompanied by strong effect sizes. This indicates that, despite the small sample size, substantial differences exist between ACH and CAH. Similarly, kinematic and kinetic parameters revealed large average differences with wide SD for both cohorts. While these variations contributed to some non-significant findings, they still indicate meaningful discrepancies in gait patterns.

A key factor influencing these variations may be the self-selected walking speed, which is known to have a substantial impact on kinematic and kinetic parameters [20]. Notably, the majority of significant differences occurred around the TO event, suggesting that this phase of the GC is a critical point of deviation between ACH and CAH. In contrast, at IC only three kinematic parameters showed significant differences, two of which were in the frontal plane. Furthermore, most deviations were observed during the stance phase and the transition into the swing phase.

Additionally, the significant differences observed at TO might be influenced by the slight delay in TO observed in ACH. Another factor to consider is the use of growth-supporting drugs among the ACH participants at the time of the study. These medications, which promote an annual growth increase of 1–5 cm, may contribute to gait deviations and could explain differences in findings compared to previous literature, particularly in relation to knee parameters.

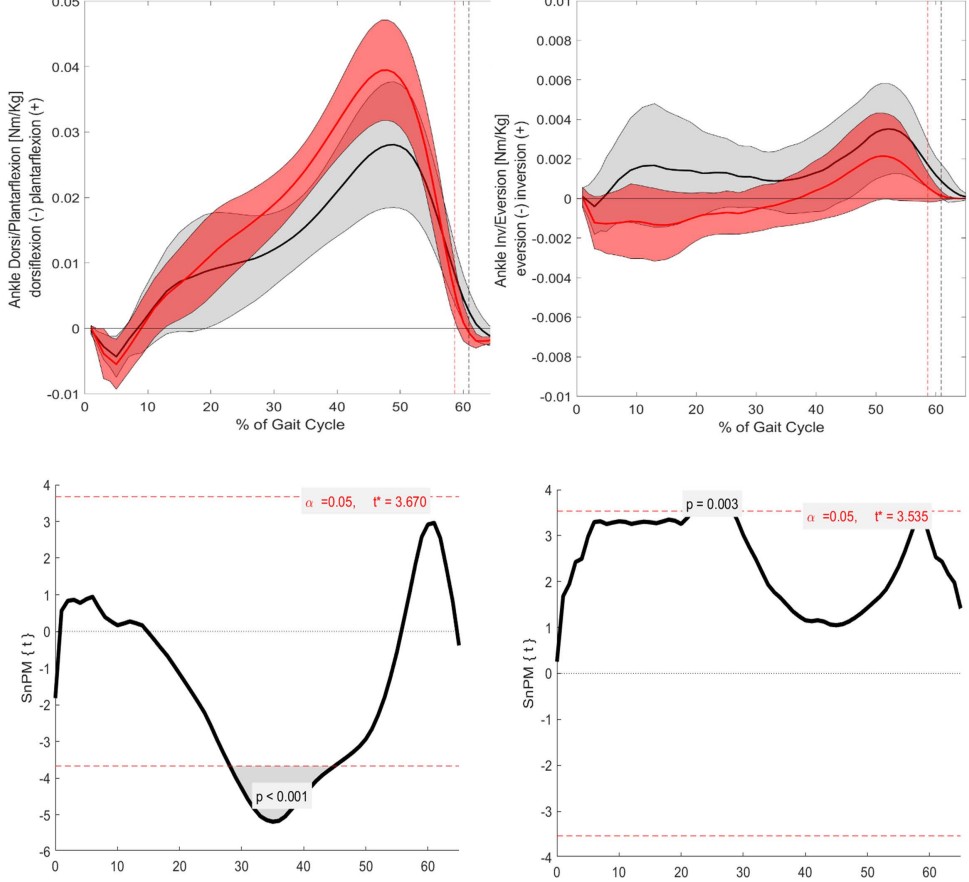

**Fig 8. Visualization of the kinetic parameters in the sagittal and frontal plane for the ankle.** Furthermore, the results of the SnPM. Alpha level set to 0.05. Black (Grey) = ACH (SD); Red (Light red) = CAH (SD). SnPM: The grey area is visualizing the significant differences.

## Limitations and future directions

This study has several limitations that should be acknowledged. First, the sample size was relatively small. Although significant efforts were made to recruit participants, the cohort size was limited by the rarity of ACH and the logistical difficulties in accessing this population. Many individuals with ACH travel long distances to attend clinical evaluations, reducing the feasibility of additional study participation. Despite this, the inclusion of 15 children within a defined age range represents a meaningful sample size. Nevertheless, the limited sample size may reduce the statistical power and generalizability of the findings. To account for this, non-parametric statistical analyses were applied.

Future research should aim to include larger, more diverse cohorts, potentially through multi-center studies or international collaborations. Increasing the sample size would allow for more robust statistical analyses and enable subgroup comparisons based on age, functional capacity, or clinical characteristics.

Secondly, the number of valid kinetic trials per participant was restricted to three per leg. This approach was based on established protocols in literature and practical constraints during data collection. The combination of short leg length and large force plate dimensions presented methodological challenges, making it difficult to obtain a higher number of valid trials without compromising data quality. Also due to the shorter legs people with ACH have the tendency to reach an earlier fatigue during walking.

Future studies should explore alternative strategies to facilitate the collection of a larger number of valid kinetic trials. This may include the use of customized force plate configurations.

Thirdly, the present study did not systematically control or analyze the influence of walking speed on biomechanical outcomes. As walking speed is known to affect kinematic and kinetic parameters, it remains unclear whether the observed alterations are partly attributable to variations in self-selected walking speeds.

Future work should investigate the effects of different walking speeds on gait biomechanics in individuals with ACH. In this study already a large range (0.67–1.32 m/s) were present, indicating a large variance in this cohort.

Another methodological limitation concerns the estimation of the HJC. The current study employed standard marker-based protocols, which may not accurately represent the anatomical HJC in individuals with ACH due to their unique skeletal structure. Previous research has demonstrated that individualized algorithms can improve the accuracy of HJC estimation in this population[22]   [23].

Future studies should consider the implementation of subject-specific algorithms for HJC calculation. Although the development and application of such algorithms are technically demanding, they may improve the precision of kinematic analyses, particularly in the transverse plane.

Despite these limitations, the findings of this study have potential clinical relevance. The identification of spatiotemporal alterations in gait patterns may inform the development of physiotherapeutic interventions tailored to the functional abilities of individuals with ACH. Moreover, the flexion patterns observed warrant further interdisciplinary investigation to clarify their biomechanical and neuromuscular origins.

Future research should prioritize the translation of biomechanical findings into clinical practice. The present results demonstrate substantial variability in the biomechanical data of individuals with ACH, as evidenced by the standard deviations observed in the graphs. This variability underscores the individuality and heterogeneity of biomechanical characteristics within the ACH population. To address this complexity, interdisciplinary collaboration among biomechanists, clinicians, and rehabilitation specialists is essential. Such collaborative efforts are critical for developing personalized treatment strategies aimed at improving functional outcomes and quality of life in individuals with ACH.

## Conclusion

The results of this study address the research questions by revealing significant differences in spatio-temporal parameters, as well as in kinematic and kinetic parameters, particularly around the TO phase of the GC. The known flexion pattern of ACH occurs for the hip and the ankle at both desired points (IC and TO). Using the results can help to develop physical therapies and interventions for ACH that prioritize the moments of the IC and TO during walking to provide a more effective walking pattern for those children.

## Supporting information

**S1 Table.**
Treatment Duration of ACH

**S2 Data.**
 Detailed Anthropometric and Gait Parameters (Averaged) of Study Participants

## Acknowledgments

The authors of this work would like to acknowledge all the children and their families who joined this study.

## Author contributions

**Conceptualization:** Mareike Hergenröther.

Data curation: Mareike Hergenröther.

Formal analysis: Mareike Hergenröther.

Investigation: Mareike Hergenröther.

Methodology: Mareike Hergenröther.

Project administration: Mareike Hergenröther.

Resources: Katja Palm, Klaus Mohnike.

Software: Mareike Hergenröther.

Supervision: Katja Palm, Klaus Mohnike, Kerstin Witte.

Visualization: Mareike Hergenröther.

Writing – original draft: Mareike Hergenröther.

Writing – review & editing: Katja Palm, Klaus Mohnike, Kerstin Witte.

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
