## [Decision Letter · Decision Letter 0]

Dear Dr.  Hergenröther,

We look forward to receiving your revised manuscript.

Kind regards,

Monika Błaszczyszyn

Academic Editor

PLOS ONE

Journal Requirements:

3. We note that your Data Availability Statement is currently as follows: “All relevant data are within the manuscript and in Supporting Information files.”

5. We note you have included a table to which you do not refer in the text of your manuscript. Please ensure that you refer to Table 3 in your text; if accepted, production will need this reference to link the reader to the Table.

Reviewers' comments:

Reviewer's Responses to Questions

**Comments to the Author**

1. Is the manuscript technically sound, and do the data support the conclusions?

Reviewer #1: Yes

Reviewer #2: Partly

2. Has the statistical analysis been performed appropriately and rigorously?

Reviewer #1: I Don't Know

Reviewer #2: No

3. Have the authors made all data underlying the findings in their manuscript fully available?

Reviewer #1: Yes

Reviewer #2: Yes

4. Is the manuscript presented in an intelligible fashion and written in standard English?

Reviewer #1: Yes

Reviewer #2: No

Reviewer #1: This paper presents a significant contribution to the field by exploring gait parameters in a unique population of children with achondroplasia, addressing an important gap in the literature and providing valuable insights for both clinical practice and future research. The following comments are intended to help improving the paper

• Abstract

• The hypothesis regarding the influence of foot-leg ratio could be elaborated for clarity.

• Statistical results should specify the significance levels for better understanding.

• The conclusion could more explicitly connect findings to potential therapeutic implications.

• Please mention the study design in the abstract or elsewhere in the study.

• A full stop is lacking in line number 26.

• The abstract is verbose at some points and would benefit from more concise language.

• Introduction

• I didn’t understand the number 25.-30.000 in first line. Is the hyphen a typo?

• Clearly articulate your hypotheses regarding the impact of foot-leg ratio on gait parameters; consider providing a brief rationale. Moreover, for better understanding of achondroplasia ‘s anthropometry, please mention other important ratios such as femur-tibia and their possible impact on gait biomechanics.

• More explicitly discuss how your findings could inform future therapies or interventions for children with achondroplasia.

• Methods

• While the sample size is stated, there is no justification provided for choosing 15 participants in each group. A discussion on how this number was determined (e.g., power analysis) would strengthen the methodology.

• The paper would benefit from mentioning the proportion of the source population from which the patients are derived.

• The recruitment process for the control group via local schools could introduce bias if certain demographics are overrepresented. More detail on how schools were selected and how representative they are of the general population would be beneficial.

• The reference on line 73 and 90 are not found. What does reference number 10 refers to? Make sure the referencing format is correct.

• The methods section could elaborate on how consistency in marker placement was ensured across all participants, as variations can significantly affect kinematic data.

• While some details about data processing are provided, further clarification on how outliers or invalid trials were handled would enhance transparency. Additionally, specifying which outcomes were analysed would provide clarity.

• Results

• Some statistical results lack complete context or clarity. For instance, when reporting significant differences in spatiotemporal parameters, it would be helpful to explicitly state which groups are being compared for each metric to avoid confusion.

• While inferential statistics are presented, there is a lack of descriptive statistics for key parameters in the results section itself. Including means and standard deviations for each group would provide a clearer picture of the data distribution. Tables of results including mean, standard deviations and effect sizes are missing.

• Discussion

• A full stop is lacking in line number 210 after reference number (21).

• The discussion is somewhat verbose and could benefit from more concise language. Some sentences are lengthy and complex, which may hinder reader comprehension. Simplifying language and breaking down complex ideas into shorter sentences could enhance clarity.

• There are instances of repetitive statements regarding findings that could be streamlined to avoid redundancy. For example, similar points about differences in gait parameters are reiterated in multiple sections without adding new insights.

• While statistical significance is mentioned throughout, there could be more emphasis on the practical significance of these findings. Discussing effect sizes in relation to clinical relevance would provide a more comprehensive understanding of the implications of the results.

• The discussion could benefit from a section on future research directions or implications for clinical practice based on the findings. This would help guide subsequent studies or interventions aimed at improving gait in children with achondroplasia.

• The conclusion can benefit from being re-written. There is no need for a long aim at the beginning and last two words of the conclusion are better to be merged and written more concise and academically.

Reviewer #2: SUMMARY

This study compared spatiotemporal, kinematic and kinetic parameters between children with achondroplasia (N=15) and an age-matched (6-12y) control group (N=15). The study found group significant differences in all outcomes except cadence. Kinematic differences were observed around initial contact for the pelvic, hip, and ankle, as well as kinetic parameters of the hip and knee around toe-off.

GENERAL/OVERALL COMMENTS & CONCERNS

Please have all text reviewed for English language and scientific style. There were many instances that the scientific message was unclear because of the current writing style. Additionally, there needs to be a clear justification for what the gap is in the literature and why it is important scientifically or clinically.

SPECIFIC COMMENTS

TITLE & ABSTRACT

Remove “in the age range of 6 to 12 years” from the title

INTRODUCTION

See above regarding greater justification for this study. E.g., previous studies must have used peak or average angle over a gait cycle or something similar & this study will look at different phases of the cycle. But how will this be used clinically? Why will this be more informative?

METHODS

Add why transverse plane kinematics of the pelvis, hip, and knee were not compared or else add.

Tell the reader if there is typically any asymmetry between sides of the body. If there is minimal/no asymmetry, then explain what data were used (e.g., a random side per person? Average of both sides?).

The authors did a great job of presenting effect sizes, but they can do better in USING their interpretation to determine significance rather than only using p-value (which is highly skewed by sample size). For instance, the authors say r =0.34 wasn’t significant because it was p = 0.06, but as defined earlier, this is a medium effect size.

RESULTS

Table 2 – clarify what “MW” means. If the Broström et al. 2022 presents the range of the SD, add that instead so that results can be easily compared to the current study.

Supplemental Table – Consider using m or f for male or female. Use footnotes to specify their meaning.

The Methods said foot-leg ratio and biomechanical variables were going to be presented, but I don’t think I saw them. Remove that aim or add data.

**Do you want your identity to be public for this peer review?** For information about this choice, including consent withdrawal, please see our Privacy Policy

Reviewer #1: **Yes: ** Fateme Khorramroo

Reviewer #2: No

---

## [Author Response · Author response to Decision Letter 1]

11 Feb 2025

I addded in the attached files a word document which all responses to the authors. I hope that is okay.

---

## [Decision Letter · Decision Letter 1]

We look forward to receiving your revised manuscript.

Kind regards,

Monika Błaszczyszyn

Academic Editor

PLOS ONE

Reviewers' comments:

Reviewer's Responses to Questions

**Comments to the Author**

Reviewer #1: All comments have been addressed

Reviewer #2: (No Response)

2. Is the manuscript technically sound, and do the data support the conclusions?

Reviewer #1: Yes

Reviewer #2: Partly

3. Has the statistical analysis been performed appropriately and rigorously?

Reviewer #1: I Don't Know

Reviewer #2: Yes

4. Have the authors made all data underlying the findings in their manuscript fully available?

Reviewer #1: Yes

Reviewer #2: Yes

5. Is the manuscript presented in an intelligible fashion and written in standard English?

Reviewer #1: No

Reviewer #2: No

Reviewer #1: The manuscript is improved. However, I think it can still benefit from a review of tone and language.

Results

There is no need for stating every report as (great), (medium). Only reporting it in methods section was enough

Limitation and future direction section can benefit from being merged and expanded, and having a more structured pattern. Stating each limitation and future suggestions consequently with enhance readability.

Reviewer #2: OVERALL

Thank you for addressing the questions from reviewers. Please have the paper re-reviewed for English. There were a few instances were this could be improved (e.g., “is without significant differences” is typically stated as “not statistically significant”; “stand phase” should be “stance phase”).

Line numbers are referenced based on the manuscript with yellow highlights.

ABSTRACT

Line 30-32: I think the assumption that understanding these differences can influence all the outcomes mentioned is a little overreaching. Can none of those interventions be designed now? How specifically can knowing kinematics, kinetics, and spatiotemporal data from gait analysis help this? I’m not certain it can, so it may be best to just delete this or be very specific.

INTRODUCTION

Line 34: Add a numerator for the prevalence of the condition (e.g., 1 birth out of every 25,000-35,000).

Line 72-73: Should the last part of this be omitted, since, as the other reviewer stated, femur-tibia ratio also differs compared to neurotypical peers “…due to the foot-leg ratio of ACH.” Or could it be caused by other factors not investigated (e.g., muscle synergies, joint range of motion, pain, ligamentous laxity)? The study design and analysis does not lend itself to being able to answer that the foot-leg ratio is the cause of gait abnormalities.

METHODS

Line 97: define BMI-SDS

Line 100: Does “supervisor” mean physical therapist or kinesiologist or someone else with specific training on marker placement?

Line 115: Add that pelvic kinematics were assessed.

Line 134: paralysis should be ‘palsy’

Line 138: edit the English, please

Line 143: “great effect” is more commonly called “large effect” in English. Update throughout the rest of the paper.

RESULTS

Table 2: please verify your calculation of normalize walking speed. For CAH, the approximate values likely fall around 0.36-0.50 (Schwartz et al., 2008 & the Brostrom et al., 2022 studies cited). Also verify normalize cadence, as the magnitude is also significantly different than these two studies.

DISCUSSION

Line 353-355: provide a reference to support the statement that strengthening interventions could reduce possible incorrect load of the joints or delete. Larger loads are inherently bad – for instance, running and squatting will results in greater joint loads than walking, but it does not mean that those loads are bad based on magnitude alone.

Line 365 and 388-390: provide more details on how these data can specifically inform physiotherapy, or delete. How do the authors define “more effective walking pattern”? Their walking pattern gets them from point A to point B, so it’s effective for them and a way for them to be independent and participate in society.

**Do you want your identity to be public for this peer review?** For information about this choice, including consent withdrawal, please see our Privacy Policy

Reviewer #1: No

Reviewer #2: No

---

## [Decision Letter · Decision Letter 2]

Differences in gait parameters between children with achondroplasia and an age-matched control group of typically developed children

PONE-D-24-44187R2

Dear Dr. Hergenröther,

We’re pleased to inform you that your manuscript has been judged scientifically suitable for publication and will be formally accepted for publication once it meets all outstanding technical requirements.

Kind regards,

Monika Błaszczyszyn

Academic Editor

PLOS ONE

Additional Editor Comments (optional):

Please adapt the final version of the manuscript to the reviewers' comments included in this message.

Reviewers' comments:

Reviewer's Responses to Questions

**Comments to the Author**

Reviewer #1: (No Response)

Reviewer #2: (No Response)

2. Is the manuscript technically sound, and do the data support the conclusions?

Reviewer #1: Yes

Reviewer #2: Partly

3. Has the statistical analysis been performed appropriately and rigorously?

Reviewer #1: I Don't Know

Reviewer #2: No

4. Have the authors made all data underlying the findings in their manuscript fully available?

Reviewer #1: Yes

Reviewer #2: Yes

5. Is the manuscript presented in an intelligible fashion and written in standard English?

Reviewer #1: Yes

Reviewer #2: Yes

Reviewer #1: My comments are attached in word file named: Achondroplasia reviewer1

The manuscript is acceptable for me after the following revision:

The phrase “Error! Reference source not found.).” appears in two spots in the introduction. Please solve the reason:

• 30 children (15 ACH & 15 CAH) in the age range of 6 to 12 years participated in this study (Error! Reference source not found.).

• Data collection took place at the sports department of the authors university using a 13-camera Vicon System with a sample frequency of 200 Hz. Age (years), height (cm), body mass (kg), sitting height (cm), BMI-SDS and foot-leg- and standing-sitting ratio were recorded (Error! Reference source not found.).

And there are still some language errors. You can use the table below to correct the mistakes:

Language Corrections for Manuscript

Original Corrected

…occurs in every 25.-30.000 births… …occurs in approximately 1 in 25,000–30,000 births…

…desired to be narrowed… …aimed to narrow the broad age range…

…the consent of participants were taken written… …written informed consent was obtained…

…hypothesized to might be caused… …hypothesized to be caused…

ACH children are under regular care… Children with ACH were under regular care…

…are causing unwanted load on the joints… …cause unwanted loading on the joints…

…at their self-selected walking speed. …at a self-selected walking speed.

It was desired to get three valid kinetic trials… Three valid kinetic trials were targeted…

…spatio-temporal, kinematic and kinetic parameters… …spatio-temporal, kinematic, and kinetic parameters…

…had significant differences, except the parameter cadence. …differed significantly, except for cadence.

…which are decisive for non-significant but great differences. …which likely contributed to non-significant findings despite notable mean differences.

…at specific points of the gait cycle gait deviations that are related to the foot-leg ratio. …at specific points of the gait cycle, deviations related to the foot-leg ratio were observed.

…a tendency to a greater abduction movement… …a tendency toward greater hip abduction…

…not easy to access this cohort and motivate them to join for additional studies. …challenging to recruit this cohort for additional studies.

Reviewer #2: OVERALL

Thank you for addressing the questions from reviewers.

INTRODUCTION

Line 72: add the words ‘which may aid in’ after gait patterns, “…gait patterns, aiding the development of…”

METHODS

Line 100: Thank you for adding the equation. I should have made my prior request more clear. Please spell out BMI-SDS here since it is the first time mentioned (e.g., I assume BMI is body mass index, but I have no clue what SDS stands for).

Line 104-5: Include what LMS stands for. And if SDSLMS is the same thing as BMI-SDS, omit SDSLMS and put BMI-SDS on the left side of the equation instead. Otherwise, it’s confusing to the reader.

Line 109: Thank you for clarifying what was meant by ‘supervisor,’ Change “supervisor” to “experienced researcher”.

RESULTS

Table 1 – clarify what the effect size is – e.g., is it a Cohen’s d for between-group differences? At first based on Lines 150-2, I thought it’d be the Pearson correlation coefficient between foot-to-leg ratio and each of the other parameters, but that doesn’t make sense because there is an effect size listed for the row “Foot-Leg Ration (%)”. If so, make it clear in the Methods how effect sizes for Table 1 were calculated. And then in the Results, if Pearson correlations were performed, present those results.

Table 2 – shows p = 0.002 for normalized cadence group differences, but Line 164 says 0.02. Please rectify the differences. Also change 0.000 in the table to <0.001.

Table 2 – spell out or include as a footnote what “MW” and “SD” mean.

Table 2 – Brostrom et al. 2022 values for normalized cadence are off by almost 100-fold from the current study. If Brostrom normalized values with a different formula, state that. The authors partly allude to this in Lines 229-232, but it still seems most likely calculation errors were made by someone or different formulas were applied.

DISCUSSION

Line 366: There is no evidence from this study that support this statement, “These findings underscore the importance of therapeutic interventions that focus on strengthening…” Delete.

**Do you want your identity to be public for this peer review?** For information about this choice, including consent withdrawal, please see our Privacy Policy

Reviewer #1: **Yes: ** Fateme Khorramroo

Reviewer #2: No

---

## [Editor Report · Acceptance letter]

PONE-D-24-44187R2

PLOS ONE

Dear Dr. Hergenröther,

I'm pleased to inform you that your manuscript has been deemed suitable for publication in PLOS ONE. Congratulations! Your manuscript is now being handed over to our production team.

Kind regards,

on behalf of

Dr. Monika Błaszczyszyn

Academic Editor

PLOS ONE